# Evaluation of Simple Algorithms for Proportional Control of Prosthetic Hands Using Intramuscular Electromyography

**DOI:** 10.3390/s22135054

**Published:** 2022-07-05

**Authors:** Nebojsa Malesevic, Anders Björkman, Gert S. Andersson, Christian Cipriani, Christian Antfolk

**Affiliations:** 1Department of Biomedical Engineering, Faculty of Engineering, Lund University, 223 63 Lund, Sweden; 2Department of Hand Surgery, Institute of Clinical Sciences, Sahlgrenska Academy, Sahlgrenska University Hospital, University of Gothenburg, 402 33 Gothenburg, Sweden; anders.bjorkman@med.lu.se; 3Department of Clinical Neurophysiology, Skåne University Hospital, 223 63 Lund, Sweden; g.andersson.lund@gmail.com; 4Department of Clinical Sciences in Lund—Neurophysiology, Lund University, 223 63 Lund, Sweden; 5The BioRobotics Institute, Scuola Superiore Sant’Anna, 56025 Pisa, Italy; christian.cipriani@santannapisa.it

**Keywords:** intramuscular electromyography, prosthetic hand control, proportional myocontrol, electromyography signal features, isometric joint force, embedded EMG processing

## Abstract

Although seemingly effortless, the control of the human hand is backed by an elaborate neuro-muscular mechanism. The end result is typically a smooth action with the precise positioning of the joints of the hand and an exerted force that can be modulated to enable precise interaction with the surroundings. Unfortunately, even the most sophisticated technology cannot replace such a comprehensive role but can offer only basic hand functionalities. This issue arises from the drawbacks of the prosthetic hand control strategies that commonly rely on surface EMG signals that contain a high level of noise, thus limiting accurate and robust multi-joint movement estimation. The use of intramuscular EMG results in higher quality signals which, in turn, lead to an improvement in prosthetic control performance. Here, we present the evaluation of fourteen common/well-known algorithms (mean absolute value, variance, slope sign change, zero crossing, Willison amplitude, waveform length, signal envelope, total signal energy, Teager energy in the time domain, Teager energy in the frequency domain, modified Teager energy, mean of signal frequencies, median of signal frequencies, and firing rate) for the direct and proportional control of a prosthetic hand. The method involves the estimation of the forces generated in the hand by using different algorithms applied to iEMG signals from our recently published database, and comparing them to the measured forces (ground truth). The results presented in this paper are intended to be used as a baseline performance metric for more advanced algorithms that will be made and tested using the same database.

## 1. Introduction

The anatomy concerning the muscles controlling the human hand demonstrates the complexity of the system responsible for the control of voluntary movement. The majority of muscles controlling hand and finger movements are located in the forearm, and they have long, sometimes multiple tendons crossing several joints, sometimes acting on several fingers. Thus, the control of hand and individual finger movements involves a complex pattern of neural activation and the inhibition of several muscles, ultimately resulting in a smooth action with the precise positioning of the joints with a force that is gradually adapted to interact with the surroundings [1].

The underlying mechanics and mechatronics in hand prostheses have been significantly improved in recent decades [2,3] and present-day hand prostheses are capable of a multitude of movements. However, most hand amputees, even if using a hand prosthesis with state-of-the-art technology, can only perform basic motor hand functions, such as opening and closing pre-selected grasps. This gap between the ability of contemporary prosthetic hands to produce dexterous movements and the actual rudimentary functionality of such hands when provided to amputees poses a challenge for researchers and companies in the field of prosthetics.

The real bottleneck preventing the optimal use of a prosthetic hand in a dexterous manner is the control part. More precisely, the methods for decoding velocities and forces of individual digits and the wrist, while interpreting the desire to move the joints of a prosthetic hand by a user, are still far from being intuitive and robust [4]. In most commercial high-end implementations, surface electromyography (sEMG) is used as a link between the amputee and the machine (prosthetic hand) [5,6,7,8,9]. However, due to the low spatial resolution and non-stationarity of sEMG, even the most sophisticated and computationally expensive algorithms are not able to extract enough information to provide robust and smooth control that resembles the behavior of the unimpaired hand. Namely, the machine learning and deep learning strategies applied on sEMG signals [10,11,12,13,14] commonly only detect the intention of an amputee to execute specific hand movements/gestures without being able to extract the desired force/velocity levels of individual joints in the hand. Therefore, the control of a multi-joint prosthetic hand usually lacks the real-time adjustment of grasps and ad hoc hand gestures that could enhance the sense of agency and function of a prosthetic hand.

Some of the control solutions that provide a higher level of functionality for prosthetic hands are oriented towards improving the quality of the signals used for control. To do so, it is necessary to obtain EMG signals close to their source, thus avoiding distortion and the loss of information that occurs due to mixing with signals from other muscles and attenuation in biological tissue. Acquiring the EMG signal close to the desired muscle has proven to yield superior results using fully implantable devices that record EMG signals directly from the surfaces of multiple muscles [15], including signals from muscles deep within the arm that are very difficult to single-out from the skin surface level. Furthermore, instead of implanting the whole recording–transmitting device, it is possible to implant only electrodes while having the associated electronics together with a power source outside the body [16,17]. This approach simplifies the surgical implantation of electrodes and the maintenance process, and the only complication is related to electrode leads failing due to excessive stress or a limited (although large) number of wire-bending cycles. With advances in surgical techniques to anchor the prosthesis using osseointegration [18,19], the longevity of the implant is further increased, thus enabling a stable and reliable interface between the human (residual muscles in the forearm) and the prosthetic hand. As a consequence of surgical and technological advances, it has become possible to derive control strategies that rely on high-quality intramuscular EMG (iEMG) signals [20] and implement them in prosthetic devices based on implantable technologies [16,21]. With a significant improvement in control signal quality, there is real potential to finally diminish one of the primary reasons for amputees’ rejection of myoelectric prostheses, which is limited controllability [22,23].

To contribute to the body of knowledge within the field of prosthetic control, our previous effort was directed towards obtaining and disseminating a database of iEMG signals [24] that could be used for developing and testing novel hand control strategies. The database uniquely combines highly selective iEMG measurements obtained using fine-wire intramuscular EMG electrodes [25] with isometric forces of individual fingers and the wrist during various hand gestures recorded using a custom-made measurement device [26]. The implicit novelty of this database is the availability of isometric hand forces elicited through a sinewave tracking task with visual feedback that then enables the evaluation of regression-based control algorithms. To provide a solid baseline of the prosthetic hand control performance, simulated offline through the utilization of the database, the present study evaluates common/well-known/simple computational methods in the scope of recorded iEMG and force signals. The methods described in the paper are utilized to show (1) the (root mean square) error between the measured and the estimated hand joints forces, and (2) the similarity of the shapes of measured and estimated forces through cross-correlation. The secondary objective of the present study is to evaluate if the common algorithms are suitable for real-time application with high sampling-rate iEMG signals. To show the relevance of common algorithms with realistic inputs, and also to serve as the reference for more computationally complex algorithms, the processing time is evaluated on various embedded and PC platforms. The purpose of the battery of tests conducted within this paper is to facilitate future studies using the same database by providing common baseline results.

## 2. Materials and Methods

### 2.1. Subjects

Fourteen male able-bodied volunteers aged between 25 and 57 years (mean 39 years) participated in the study which was divided into two protocols. As the volunteers were equally divided between protocols and two of the volunteers were included in both protocols, the final database contained sixteen files, eight for each protocol. All the volunteers signed an informed consent, and the study was approved by the Regional Ethical Review Board in Lund, Sweden (Dnr 2017-297).

### 2.2. Recorded Signals

The signals database [24] was recorded in a setup that included intramuscular fine-wire EMG electrodes in a differential recording configuration, with volunteers performing a set of hand gestures in a timely fashion and the acquisition of isometric hand forces using a custom-made measurement device.

The iEMG signals were recorded from 9 forearm muscles divided into 2 groups which were associated with the 2 levels of amputation:The short residual limb (SRL) protocol targeted the following muscles: flexor carpi radialis (FCR), extensor carpi radialis (ECR), pronator teres (PT), flexor digitorum profundus (FDP), extensor digitorum communis (EDC), and abductor pollicis longus (APL);The long residual limb (LRL) targeted the following muscles: flexor digitorum profundus (FDP), extensor digitorum communis (EDC), abductor pollicis longus (APL), flexor pollicis longus (FPL), extensor pollicis longus (EPL) and extensor indicis proprius (EIP).

The positioning of the fine-wire electrodes was performed by a senior consultant in clinical neurophysiology. The signals picked up by the pairs of fine wires (in differential recording configurations) were amplified and digitalized using the Quattrocento biomedical amplifier system from OT Bioelettronica, Torino, Italia. The amplifier had a common mode rejection ratio (CMMR) higher than 95 dB, an input resistance higher than 10^11^ Ω on the entire bandwidth, and a noise level lower than 1 µV_RMS_. To reduce the interferences induced in the leads between the fine wires and the amplifier, shielded differential preamplifiers were used in close proximity to the fine-wire skin entry points. In parallel with the iEMG recordings, forces resulting from hand muscles contractions were also recorded using the measurement device which kept the hand stationary during the procedure. In total, 9 force gauges picked up the resulting hand forces: one per finger (D2 (index finger)–D5 (little finger)), two for the thumb, and three for the wrist. The recording protocol was guided by an automated graphical user interface which, in a strictly timed manner, displayed commands and cues to the volunteer. It also provided feedback on the exerted force which was crucial for force matching/tracking tasks. An example of this functionality is the sine wave tracking task which started by instructing the volunteer to perform maximal voluntary flexion and the extension of a finger, thumb or the wrist, the adduction–abduction of the thumb, or the pronation–supination of the wrist. Although the inserted electrodes targeted only a subset of the muscles involved in the abovementioned hand movements, the protocol was consistent for all volunteers and all electrode placements. This way, it was possible to compare muscle synergies and crosstalk between iEMG channels, which is not in the focus of this study, but might be of interest for other studies that use the database. Then, the software generated a low-frequency sine wave with an amplitude equal to 20% of the maximal voluntary flexion/extension and a frequency of 0.1 Hz and displayed it in the same graph with the finger or wrist force that the volunteer produced (see Figure 1). The idea here was that the volunteer would match the computer-generated sine wave by continuously eliciting an equivalent force.

Although the measurement protocol included various hand gestures, for the scope of this paper, only the sine tracking data were used because the aim of this paper is to produce the direct and proportional control of a multi-joint prosthetic hand.

As the iEMG was recorded at 10 kHz, the force data were up-sampled for easier manipulation. In addition, the commands and cues presented to the volunteers were recorded for the offline segmentation of the measurements into individual finger/wrist movements. The Matlab data container files (*.mat) can be found here: (https://figshare.com/s/06f113bd74ecf6384729 (accessed on 2 June 2022)).

### 2.3. Signal Pre-Processing

As the methods presented in this paper are intended for online implementation, the recorded signals were minimally pre-processed. Namely, the powerline noise was superimposed with the iEMG signal, which was relatively weak compared to the sEMG. To remove the powerline base frequency (*f_0_*) and its harmonics (*f_0_ × i, i = 2, 3, 4*…), which also exceeded iEMG signal amplitude, a band-stop comb filter comprising third-order Butterworth notch filters (*f_0_* = 50 Hz and Δ*f* = ± 2 Hz) was applied to the recorded signals [27]. The next pre-processing step was the segmentation of recorded signals based on the automatic data labeling performed during the recordings. In this stage, force outputs were divided between positive and negative values or phases (flexions–extensions, adductions–abductions, pronations–supinations) so that they could be independently associated with different muscles. Finally, as the last step preceding the algorithm stage, optimal matchings between targeted muscles, elicited movements and the force gauges were established. On top of that, within the selected force channel, a phase (positive or negative) was associated with the actions of specific muscles. In other words, for each iEMG signal (muscle) there was an associated hand movement that specifically activated the muscle where the fine-wire electrode was placed. Furthermore, there was a sensor within the force measurement device that registered the force generated by the muscle contraction of interest. For example, activation in the PT muscle was always associated with the command to pronate the forearm and the elicited force was measured by the sensor designated for the measurement of wrist pronation. Nevertheless, for some muscles, there could be a variety of different matchings for different subjects due to the multi-compartment nature of the muscles in which the fine-wire electrodes were anchored or the individuals’ strategies for producing the commanded contractions. Such examples include the FDP, which could flex any of the D3–D5 fingers depending on the activated compartment, or EDC, which extended the D2–D4 fingers. Furthermore, depending on individuals’ synergies, APL could be more active during the command to push the thumb to the left (right-handed setup) or upwards. An example of the signal pre-processing method is shown in Figure 2.

The iEMG–movement–force matchings were initially selected based on the signal-to-noise ratio of iEMG within different segments (movements) and refined based on the minimal root mean square error (RMSE) between a normalized force gauge output and the estimated force using the MAV algorithm.

### 2.4. Testing of Algorithms

The algorithm evaluation procedure was designed with a specific prosthesis-use scenario. The performance of the algorithms was tested in a similar fashion as that shown in Figure 2 by using the optimal pairings between the iEMG channels, the phases of the force channels, and the movements found during the signal pre-processing stage. It was envisioned that the full setup should be as quick as possible. In line with this principle, only the first two of ten sine tracking periods per tracking task were selected to normalize the outputs of the algorithm for force estimations (similar to the previous study [28]), while the following tracking periods were used only for evaluating the performances of the algorithms. Such a condition results in fast calibration that requires only two repetitions of slowly increasing and decreasing isometric force elicited by muscles during various hand movements. Although in the case of amputees there is no possibility of tracking muscle forces, the same principle still applies as the only purpose of the initial calibration is to match muscle activity and forces, and to normalize force output. Thus, in the realistic scenario, the amputee will be asked to slowly increase and decrease force while imagining specific hand movements [29].

The testing of the iEMG algorithms for extracting signal features was also performed in a manner that corresponded to the real-time application. To achieve this requirement, all algorithms were implemented within the moving/sliding windows of different widths where the computed value was adjoined with the last (newest) sample from the window. This scenario corresponds to the direct control of a prosthesis degree of freedom in real-time using a single muscle.

The algorithms selected for evaluation are already well-known from numerous publications in the field of prosthetics [30,31,32]. Due to relatively poor sEMG signal quality, these algorithms are often used only for feature extraction, while some machine learning algorithms are used to extract relevant information regarding muscle activation. The following algorithms were systematically used in conjunction with the recorded iEMG signals and evaluated against measured forces.

1.Mean Absolute Value

Mean absolute value (MAV) is the favored EMG feature in many myoelectric control applications [33,34]. It is calculated within a moving average window of the rectified iEMG signal.

2.Variance

Variance (Var) is the mean value of the square of the deviation of the signal [35]. As the mean of iEMG is close to zero, the equation can be simplified for faster calculations.

3.Slope Sign Change

Slope sign change (SSC) is a time-domain method used to estimate the frequency feature of the iEMG signal [36]. The calculation of the SSC relies on counting changes between positive and negative slopes among three consecutive samples. To limit SSC calculation only to periods with iEMG activity, the threshold function is imposed in the feature extraction method.

4.Zero Crossing

Zero crossing (ZC) is the function that counts the number of consecutive EMG samples that change signs within the sliding window [33]. Similarly to the SSC calculation, the threshold function is imposed to remove the calculation of ZC during periods without pronounced EMG activity.

5.Willison Amplitude

Willison amplitude (WA) is a measure related to the superimposed action potentials that make the EMG signal [37]. The WA is the number of consecutive differences between consecutive samples that exceed the set threshold.

6.Waveform Length

Waveform length (WL) is the cumulative length of the waveform over the time window [38].

7.Envelope

The signal envelope (ENV) is calculated as the root mean square over the time window.

8.Total signal energy

The total signal energy (Etot) is defined as the sum of spectral amplitudes calculated using fast Fourier transformation.

9.Teager energy in time domain

The Teager energy operator is a non-linear operator that can track the energy and identify the instantaneous frequencies and instantaneous amplitudes of signals at any instant. It strongly correlates with the nature of the iEMG signal which comprises individual action potentials [39,40,41]. Teager energy in the time domain (Ttd) was calculated using the Teager–Kaiser energy operator. The resulting values were then smoothed using a moving average filter defined in the same manner as the sliding windows present in other algorithms.

10.Teager energy in frequency domain

Another implementation of the Teager energy operator is in the frequency domain (Tf) using fast Fourier transformation [42].

11.Modified Teager energy

The modified Teager energy (Tf_mod) is similar to the Teager energy operator in the frequency domain (Tf) with the only difference being that the frequencies are not squared.

12.Mean of signal frequencies

Mean of signal frequencies (MNF) is calculated from the windowed iEMG power spectrum as the average frequency [43].

13.Median of signal frequencies

Median of signal frequencies (MDF) is calculated as the frequency at which the power spectrum is divided into two regions with equal power [43].

14.Firing rate

The calculation for the firing rate (FR) iEMG signal feature is performed by implementing the algorithm presented in [21]. The algorithm is based on counts of action potentials within the iEMG signal that are larger than the set threshold.

The mathematical formulas used to compute algorithms are given in Table 1.

To provide greater insight into the iEMG signal properties, the abovementioned computational methods were optimized to provide the lowest discrepancies between measured and estimated forces. The optimization (parameter exploration) was performed in a brute-force manner by testing all combinations of parameters within defined ranges. The rationale behind this method is to provide outputs for the incrementally increasing parameters in order to obtain the gradual evolution of error between measured and estimated force. The main reason for having such data is to identify trade-offs between force estimation error and computational complexity.

The computational complexity of individual algorithms is estimated theoretically and practically. For the theoretical computational complexity, which is completed in big O notation, the algorithms were analyzed down to the basic arithmetic operations. The results for theoretical computational complexity are given in Table 2. The practical computational complexity of the algorithms was evaluated as the execution time for different window sizes. To ensure that the processing was not biased by the specific processor architecture, the tests were repeated on different platforms:PC with Intel i7-6700K, 64-bit processor at 4 GHz;Teensy 4.0 with Cortex-M7, 32-bit processor at 600 MHz;Teensy 3.6 with Cortex-M4, 32-bit processor at 180 MHz;BLE-Nano with Cortex-M0 (NRF51822), 32-bit processor at 16 MHz;Arduino-Nano with ATmega328, 8-bit processor at 16 MHz.

The implementation on the PC platform was performed with Matlab 2018b (The MathWorks, Natick, MA, USA) scripts where the timing was an average of all sliding window computations within the whole signal. It should be noted that the execution time of algorithms on the PC platform was not deterministic nor constant, as the operating system (Windows 10, Microsoft Corporation, Redmond, WA, USA) was not in real time. To minimize the variability of the execution time required for single window processing, the results were based on the averaged values for all signals and all subjects. Thus, the execution times on the PC platform are given only for inter-algorithm comparison. The implementations on other real-time platforms were written and compiled in the Arduino IDE environment using a single-precision floating-point format (float32) which guarantees the same conditions for all three processors. All functions, except for fast Fourier transformation (FFT), were written using basic mathematical operations. The Arduino library (github.com/kosme/arduinoFFT (accessed on 18 February 2022)) was modified for single precision and used as an FFT function in Arduino IDE. The exact duration of individual algorithm executions was measured by pulling a single digital output port during the computations. Furthermore, for the processors based on the Cortex-M4 or M7 architecture, some of the algorithms were implemented using CMSIS-DSP functions that are highly optimized for these types of operations.

The parameter included in all the algorithms that was varied was window size. An additional parameter present in some of the algorithms was the threshold (SSC, ZC, WA, and FR) which was also varied together with the window size. The list of varied parameters is shown in Table 2.

The range of window sizes tested within this study was 50–1050 ms, which is wider than that of the state of the art [32,44,45,46]. There are several reasons for extending the range of tested window width. First, we wanted to provide computational/practical evidence of an optimal range of window sizes that should be used in conjunction with iEMG signals. Secondly, a wider window impacts computational time, which in turn results in increased controller delay, which has been identified as one of the obstacles in providing intuitive prosthetic control [9,47]. As microprocessor technology is continuously advancing, there is a need to update general knowledge regarding computational time for different EMG feature extraction algorithms, window widths, and processor architectures. Thus, providing both quantitative and qualitative information about algorithm performances, even for wide window sizes, should be beneficial for future studies. The increment of the sliding window width for all algorithms was set to 100 ms to provide 10 equally spaced evaluation points. The thresholds for SSC, ZC, and WA were defined as signal “dead zones” and, as in [33], were based on the system noise. In this study, we estimated the noise by calculating the MAV of the EMG during the rest period. The MAV (rest) was computed from the first 100 ms of each segmented movement (isometric contraction) which did not contain voluntary iEMG, as the command for initiating muscle contraction was delayed in order to enable the subject to focus on the subsequent hand movement. For the purpose of estimating the optimal dead zone range, thresholds between 0 and Q × MAV were used, where Q was the multiplier of the MAV (rest) value. The parameter Q was varied in the range (0–4) with 0.2 increments during the exploration of the parameters. The threshold for the FR algorithm was calculated as a quantile of the iEMG signal during the first full period of the sine tracking task and it varied between the 85th and 99th quantile with 1 quantile increments.

### 2.5. Evaluation

The outputs of the algorithms were evaluated with respect to the elicited force which served as the ground truth. As previously stated, both estimated and measured forces were normalized using only the first two full periods of the sine tracking task. The metrics selected for the evaluation of the algorithms included root mean square error (RMSE) and cross-correlation (Pearson), as in [28]. As the data used to compare different algorithms and window sizes did not follow the normal distribution determined by the Lilliefors test, we used the Friedman test with Bonferroni post hoc correction to assess the statistically significant differences of the medians.

## 3. Results

A sample of the measured and estimated forces is shown in Figure 3.

In addition to the theoretically calculated computational complexity of each algorithm, the first performance parameter of the tested algorithms was the execution time across different platforms. These results are presented in Table 3. As mentioned before, execution times on the PC platform were influenced by the implementation in Matlab, while the execution times on other platforms could be directly compared as they used the same source codes.

As the aim of this paper is to evaluate the possibility of direct force tracking using only iEMG signals, the most appropriate performance evaluation metric was the root mean square error (RMSE) between the measured and the estimated force signals. This was also the most rigorous metric, as it showed sample-to-sample difference without any time-lag compensation between the measured and estimated joint forces. The RMSE results for each finger movement and each subject (6 iEMG channels × 16 datasets) were aggregated to enable stronger statistical analysis (see Figure 4).

As the cumulative RMSE data did not follow a normal distribution, the Friedman test with Bonferroni correction for multiple comparisons was used to assess the differences between the algorithms. The results of the statistical analysis are summarized in Table 4. Most notably, there was a highly statistically significant difference (*p* < 10^−6^) between the mean ranks of MNF, MDF and FR and the rest of the algorithms. In addition, other statistical differences could also be found between some of the other algorithms, as shown in Table 4. This table was derived from the best performances of each algorithm and for each signal, regardless of the algorithm parameters (such as window size).

The cross-correlations between the measured and estimated forces are provided in Figure 5. This metric is complementary to RMSE as it quantifies the similarity of the signal shapes. Cross-correlation and RMSE provided different insights into matching between measured and estimated forces, and the statistical analysis showed different statistical differences between the algorithms.

In addition to comparing different algorithms, this paper shows the impact of parameters associated with individual algorithms on the RMSE and cross-correlation metrics. For all the algorithms, the computational sliding window size was used as a variable parameter. The range for this parameter was broader than in other relevant publications [48]—between 50 ms and 1050 ms—so as to obtain optimal values and assess the algorithms’ performances in extreme setups. Furthermore, some of the algorithms were based on two adjustable parameters: window size and a threshold. Thus, during the algorithm evaluation, both of the parameters were independently varied within the broad range of values. Figure 6 depicts the impact of the variable window size on the MAV algorithm metrics, but a similar behavior could be seen in all of the one-parameter algorithms (see Appendix A section). The MAV algorithm was chosen as an example, as it is commonly used in myoelectric applications [30].

In the case of two-parameter algorithms (SSC, ZC and FR), the second parameter is the threshold. SSC and ZC use a threshold to exclude events within a dead zone, while FR relies on the signal amplitude percentile-based threshold to detect motor units firing above it (above the noise level). Figure 7 and Figure 8 show the RMSE and cross-correlations for the SSC and FR algorithms when two parameters were varied. A figure depicting the performance of ZC during the two-parameter sweep is similar to SSC, and is available in the Appendix A section. The results presented in Figure 7 show that, in the case of FR, both parameters influence the performance of the algorithm, while SSC (and ZC) is more sensitive to threshold changes (as in Figure 8).

## 4. Discussion

The central part of the paper is the database of iEMG signals that we have recently published [24] that are suitable for developing and testing novel algorithms for prosthetic control. Thus, to make upcoming methods developed using our database comparable, there is a need to establish a golden standard based on simple, well-established computational methods. For this purpose, the set of common algorithms/methods that are usually utilized only for EMG feature extraction was selected. However, instead of only being used the first stage of a prosthesis control scheme, in the scope of this study, they were elevated to present the sole link between muscle activation and prosthetic hand actuators. It has to be noted here that the paper does not present any computational novelties per se, but its contribution is the combination of unique datasets and well-known, common feature extraction algorithms. The results presented in the paper can be divided into three categories: (1) the comparison between different algorithms in terms of RMSE and cross-correlation metrics; (2) the exploration of the influence of the algorithms’ parameters on the RMSE and cross-correlation metrics; and (3) the estimation of the computational times of algorithms for various embedded and PC platforms.

An analysis of the RMSE metric of different algorithms showed statistically significant differences between FR and all other algorithms. This clear advantage of FR is somewhat unexpected, as some of the other algorithms are commonly used, and preferred, for prosthetic control, especially in commercially available systems [4]. We hypothesize that the main reason for such performances with the FR algorithm was the high quality of iEMG recordings that selectively picked up the activity of agonist muscles during isometric force generation. In addition, with the small electrode area, only a few motor units were predominately present in the iEMG signals; thus, it was possible to estimate the firing rate of motor units with high precision, compared with the case of surface EMG signals which incorporated the firings of tens or hundreds of motor units. On the other end of the algorithm spectrum, frequency-based algorithms relying on dominant frequency within a time window (MNF and MDF) consistently underperformed compared to the other algorithms. With the iEMG signals that were used, these algorithms predominately acted as methods to detect muscle activity, with a sharp rise upon the initiation of muscle contraction and a mostly flat response during contraction. Again, we hypothesize that this behavior was due to the selective nature of the recorded iEMG samples containing sparse motor unit action potentials, compared with the superficial EMG samples which comprised superimposed action potentials. This, in the case of highly selective iEMG, indicates that mean and median signal frequency were more related to the shape of individual action potentials. On the other hand, in the case of superficial EMG, frequency measures of the signal were more related to the sheer number of the superimposed action potentials which corresponded to the muscle force. The analysis of the cross-correlation metric was less deterministic in terms of clear overperforming and underperforming algorithms. As in the case of the RMSE metric, FR generally estimated force shape better than the other algorithms, while MNF and MDF tended to distort the iEMG-based force estimation more than the other algorithms.

In addition, the optimal window size for different algorithms was also investigated in the current study. A wide range of window sizes (50–1050 ms) was used to assess the performance of the algorithms. The results indicating the best tradeoff between RMSE, correlation, and the controller delay (due to the window size) for the MAV algorithm are presented in Figure 6. There was no statistically significant difference in RMSE metrics between 250 ms and the larger windows; thus, 250 ms should be favored in the use case requiring minimal RMSE and controller latency. In the case of cross-correlation, with the increase in the window size up to 450 ms, there was a statistically significant increase in correlation metrics. As beyond this point there was no significant difference between the calculated cross-correlations between estimated and measured force, 450 ms should be used to obtain the best signal shape approximation if the MAV algorithm is to be used. Examples for other algorithms are provided in the Appendix A section.

The results of applying simple computational algorithms with this database could also be compared to other studies. For example, Kamavuako et al. [28], in their study, used a sine tracking protocol with isometric forces corresponding to two wrist DoFs, which is relatively similar to the protocol used in this paper. The force level was >10% of the MVC level, which was also the case in the study presented here. The results for the ten subjects reported by Kamavuako et al. showed a mean RMSE of ~14% (range between 10% and 19%). These values are in line with the best-performing algorithm (FR) from our study which had a median RMSE of 17.8% (Q1-13%; Q3-26%). For the other algorithms, there is partial overlap with the results of Kamavuako et al.’s study (e.g., for MAV, median RMSE was 22.5% with Q1-17%; Q3-29.6%). In terms of cross-correlation metrics, Kamavuako et al. achieved a mean Pearson correlation coefficient of ~0.93 (range between 0.88 and 0.96). The results of the presented study show a lover cross-correlation between the isometric forces and the tested simple algorithms. The best-performing algorithm (FR) only partially falls into the range reported by Kamavuako et al., with a median cross-correlation of 0.85 (Q1-0.75; Q3-0.92), while for the other algorithms, only Q3 values were above 0.88. We hypothesize that there are three potential causes of the somewhat worse results presented in this study compared with the study performed by Kamavuako et al. Firstly, the algorithms used in our study were very basic; the normalization of the outputs was based on only the first two force sine tracking repetitions, and no additional optimization was performed. Secondly, the correlations between some of the iEMG channels and the elicited joint forces were labeled as “questionable” during the qualitative evaluation of signal quality completed by the neurologist (check [24] for more information). Nevertheless, those pairings were still included in this study as they represent realistic use cases where fine wires, or intramuscular electrodes in general, were slightly misaligned with the targeted muscle, or migrated after initial insertion. Thirdly, in the presented study, only direct control was considered (one iEMG channel to estimate one joint force), while in the study completed by Kamavuako et al. eight iEMG channels were used to estimate two wrist DoFs, therefore using more information regarding muscle activity to estimate the resulting forces.

In terms of computational complexity, the approach of using simple feature extraction algorithms as the main force regression method significantly reduces the necessary processing power, which is an important step in the development of an embedded real-time prosthetic hand controller. Preferably, the controller should provide an immediate joint force output response to the user command to enhance the sense of agency of the prosthetic hand. Due to the latencies of the motorized actuator system, the computational aspect of the prosthesis control chain should have a delay below 125 ms [47] so that the total latency of the prosthetic hand reaction stays below 300 ms, which is still considered acceptable [9]. To keep the computational latency as short as possible, one of the solutions is to increase computational window overlap. Ideally, the sliding window increment of only one sample would yield the smallest computational delay. However, this principle requires computation to be performed between consecutive EMG (or iEMG) samples, which in the case of the iEMG used within this study should be completed within 100 µs. State-of-the-art prosthetic hand control methods do not use such rigorous conditions, but permit window overlaps of 50–300 ms [9,49].

In the context of the iEMG used within the scope of this paper, which was sampled at 10 kHz, the embedded implementations of MAV, Var, WA, WL, Env, and Ttd were able to perform real-time control for the window sizes found to be optimal. In other words, with window sizes up to 450 ms (4626 samples), the listed algorithms would execute computations under 0.1 ms (running on a Cortex-M7), ensuring a window increment of only a single sample. In the case of other time-domain algorithms (SSC, ZC and FR), the increment should be more than two samples (>0.2 ms), which still fulfils the commonly used computational delay requirement [47] by far. Finally, frequency-domain algorithms require more time to compute, but with FFT library optimization, it is possible to obtain updated control (sliding window increment) every 1 ms for window sizes up to 450 ms. The computational time simulations carried out on Cortex-M7 clearly showed the ability of the implemented algorithms to run in real time, even with the high sampling rate used in this study. In the case of commonly used estimation update rates (50–300 ms), most of the tested algorithms with maximal tested window sizes were calculated within this computational window. Only the FFT-based algorithms exceeded this range of acceptable delays for window sizes longer than 256 samples when computed on Cortex-M0. In the case of ATmega328, the main restriction was relatively limited RAM, which permitted tests on window sizes of 128 samples or less. With advancements in microprocessor/microcontroller technology, it is safe to assume that the computational load of calculating the algorithms evaluated within this study would not present an issue, even with extreme conditions such as low latency coupled with low power consumption.

A strong point regarding the methodology presented in this paper is the calibration phase, which tunes the output of individual algorithms to match the level of the desired joint force. This phase comprises only two repetitions of slowly increasing/decreasing self-perceived muscle contractions in the case of amputees, or isometric muscle force in the case of the signals used within this paper. Thus, the transfer function between iEMG and the force of an actuated prosthetic hand joint is governed by self-selected maximal muscle activity level. Although this method is intended for the recording/testing protocols associated with this iEMG database (ten repetitions, two for tuning and eight for performance evaluation), there are also implications for real-world prosthetic control strategies. Specifically, based on our findings, by executing one or two slow muscle contractions per controllable degree of freedom while recording iEMG, it should be possible to establish direct muscle control. Therefore, the initial calibration process could be relatively quick when compared with the machine learning methods that usually require a large number of repetitions to build an accurate movement model.

A limitation of the results presented here is related to the dataset of iEMG signals that was recorded and made publicly available [24]. Some of the findings presented in this paper might be challenged with methods based on other iEMG signal databases or real-time control approaches. Nevertheless, our intention is to provide baseline measures for the proportional/direct control of a dexterous prosthetic hand using well-known and extensively used computational methods. In this way, future studies using our iEMG database as the base for developing and testing novel algorithms could use presented results instead of independently implementing and optimizing “baseline” algorithms. Another limitation of this study is that all evaluations were completed offline. Therefore, as with any offline simulation study, the results should be taken with caution, as in real-time application there is always an interaction between controller output and user control in a closed-loop manner. Furthermore, physiological factors (such as fatigue or cognitive load) and environmental obstruction can have a significant impact on the accuracy of any online evaluated control method.

The methods presented in this study were devised in line with recent trends in prosthetic technology, namely, osseointegration [18], targeted muscle reinnervation (TMR) [46], and intramuscular EMG recording [15]. Although the combination of these techniques is currently not a clinical standard for prosthetic users, strong results of small sample studies have shown the great potential of these methods to increase the usability of hand prostheses. We assume that a combination of these techniques will become the state of the art in both research studies and clinical applications; therefore, providing an appropriate iEMG signal database alongside the evaluation of simple algorithms’ performances could be useful in future studies. Furthermore, within this study, only a subset of the available muscles were targeted by the fine-wire electrodes inserted through the skin to obtain representative signals. The combination of osseointegration and TMR could enable the placement of more intramuscular electrodes with greater precision during a surgical intervention, therefore establishing a genuine foundation for the direct/proportional control of prostheses’ individual degrees of freedom.

## Figures and Tables

**Figure 1 sensors-22-05054-f001:**
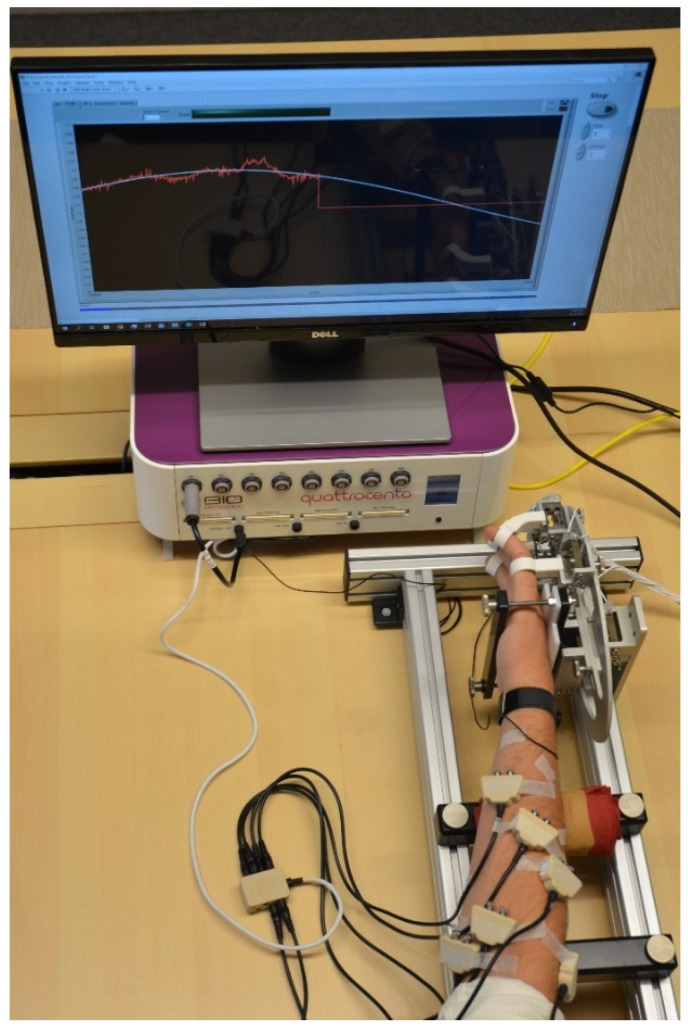
Measurement setup. The participants were following the slowly changing sine wave (repetition frequency was 0.1 Hz) with a finger DoF by producing isometric contractions. The forces were acquired by the force measurement device and displayed in real-time on the screen. Fine-wire electrodes were taped at the skin entry point to prevent accidental pulls. To reduce induced noise, the preamplifiers were placed as close as possible to the entry points of the fine wires. In addition, the wires between the entry point and preamplifiers were taped to the skin. The metal frame of the force measurement device was grounded to reduce powerline noise.

**Figure 2 sensors-22-05054-f002:**
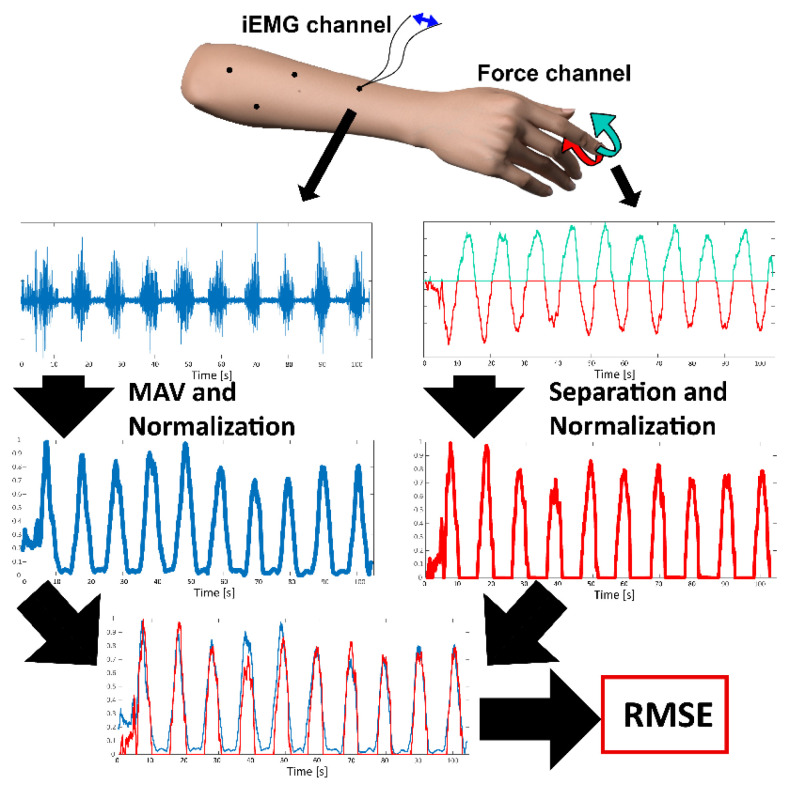
Schematic representation of the signal pre-processing and testing of algorithms steps. For each iEMG channel (left side, blue signals), the MAV feature was extracted. In the following step, the MAV features were normalized for each movement based on only two first sine tracking periods (out of 10 repetitions). Hand and wrist forces (right side) were divided into positive (teal) and negative phases (red signals), after which individual phases were normalized (0–1) for each movement. To establish a correlation between iEMG channels (differential wires) and the phases of the force channels, a systematic evaluation of RMSE metrics between normalized MAV features and forces was conducted. The result of this process was a lookup table connecting iEMG channel, force channel (gauge), force phase, and movement (see the file “Fine-wire force-emg pairs” provided in the database paper [24]). The example signals are based on subject 4’s FPL muscle and thumb flexion/extension.

**Figure 3 sensors-22-05054-f003:**
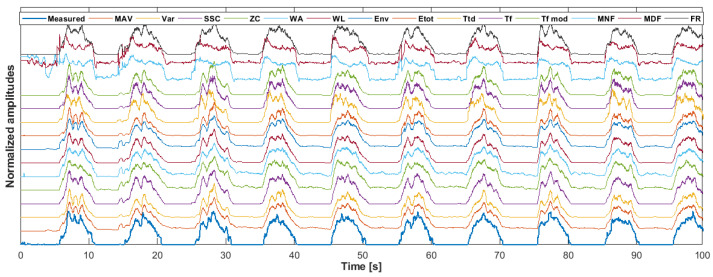
Measured force (blue—bottom row) and estimated forces using different algorithms. Each algorithm output (and the force signal) was normalized using the maximal amplitude within the first two repetitions as the scaling factor. The example signals are based on subject 15, EDC muscle and ring finger extension.

**Figure 4 sensors-22-05054-f004:**
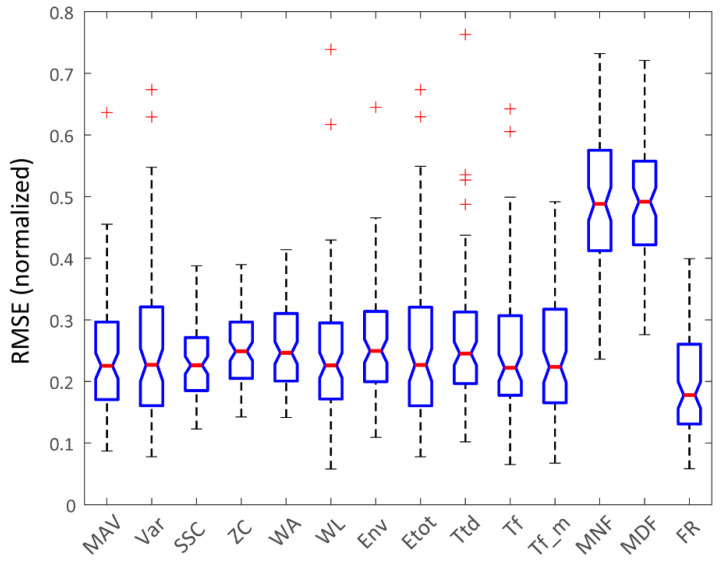
RMSE metric for all methods. These data comprise all subjects and all joint force estimates (except for the channels labeled as “poor”, see [24]).

**Figure 5 sensors-22-05054-f005:**
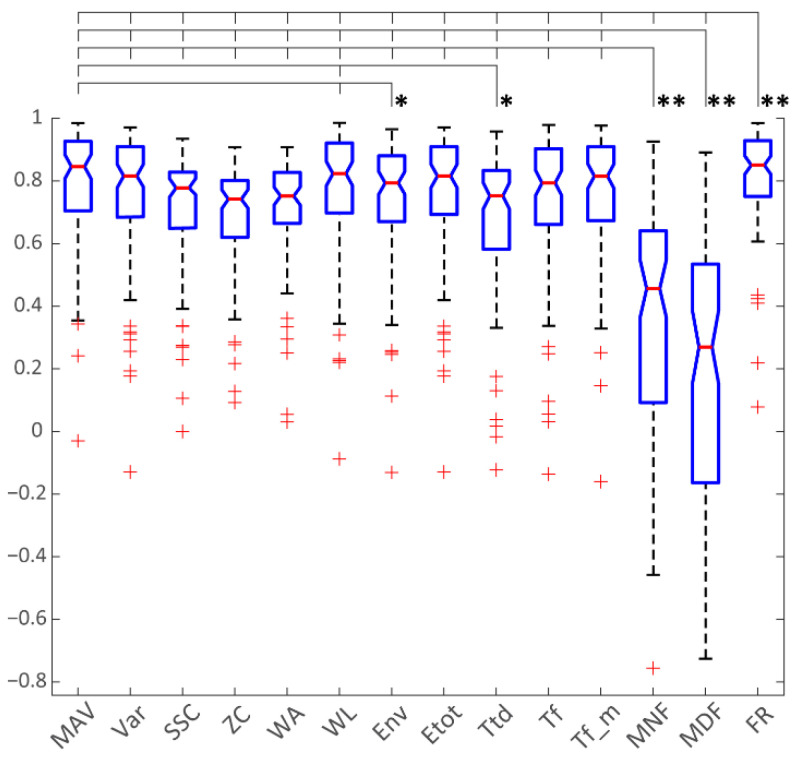
Cross-correlation between the algorithms and joint forces (ground truth). These data comprise all subjects and all joint force estimates (except for the channels labeled as “poor”). * *p* < 0.05, ** *p* < 0.01.

**Figure 6 sensors-22-05054-f006:**
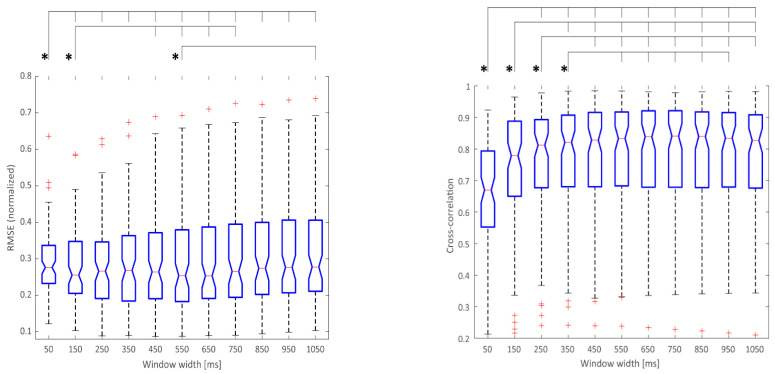
RMSE (**left**) and cross-correlation (**right**) of MAV algorithm for different window sizes. The cases of significant differences (*p* < 0.05; Bonferroni-corrected) are marked with *.

**Figure 7 sensors-22-05054-f007:**
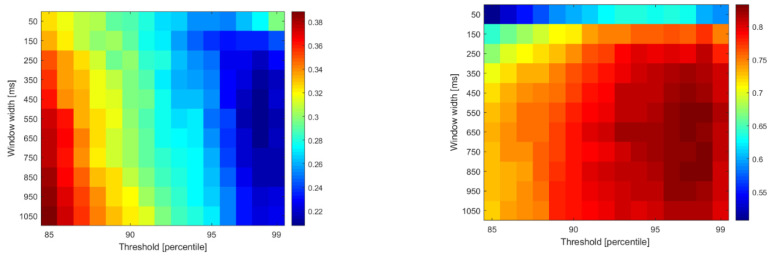
RMSE (**left**) and cross-correlation (**right**) for firing rate with different thresholds and window widths.

**Figure 8 sensors-22-05054-f008:**
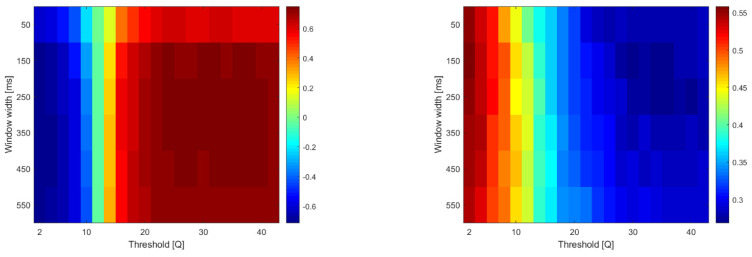
RMSE (**left**) and cross-correlation (**right**) for SSC with different thresholds (Q multipliers of MAV at rest) and window widths (actual value of Q is shown/10). Finding: with threshold = 2 × MAV (rest) the window can be short (150–250 ms).

**Table 1 sensors-22-05054-t001:** Formulas used to compute algorithms that were evaluated. N denotes window length, $x_{n}$ is the *n*-th iEMG sample within the current window, *M* is the number of frequency bands, $P_{n}$ is the spectral power of the n-th frequency band, $f_{n}$ is the central frequency of the n-th frequency band, and AP the number of detected action potentials over the threshold within the sliding window function.

	Algorithm	Acronym	Formula
**1**	Mean absolute Value	MAV	$\mathrm{MAV}=\frac{1}{N}\;\overset{N}{\underset{n=1}{\sum }}\left|x_{n}\right|$
**2**	Variance	Var	$\mathrm{Var}=\frac{1}{N-1}\overset{N}{\underset{n=1}{\sum }}x_{n}^{2}$
**3**	Slope sign change	SSC	$\mathrm{SSC}=\overset{N-1}{\underset{n=2}{\sum }}f\left[\left(x_{n}-x_{n-1}\right)\times \left(x_{n}-x_{n+1}\right)\right]$ $f\left(x\right)=\{\begin{array}{c} 1,\;\;if\;x\geq threshold\\ 0,\;\;otherwise\;\;\;\;\;\;\;\;\;\;\;\;\; \end{array}$
**4**	Zero crossing	ZC	$\mathrm{ZC}=\overset{N-1}{\underset{n=2}{\sum }}[sgn\left(x_{n}\times x_{n+1}\right)\times f\left[x_{n}-x_{n+1}\right]]$ $f\left(x\right)=\{\begin{array}{l} 1,\;\;if\;x\geq threshold\\ 0,\;\;otherwise\;\;\;\;\;\;\;\;\;\;\;\;\;\; \end{array}$
**5**	Willison amplitude	WA	$\mathrm{WA}=\overset{N-1}{\underset{n=1}{\sum }}f\left[\left|x_{n}-x_{n+1}\right|\right]$ $f\left(x\right)=\{\begin{array}{l} 1,\;\;if\;x\geq threshold\\ 0,\;\;otherwise\;\;\;\;\;\;\;\;\;\;\;\;\;\; \end{array}$
**6**	Waveform length	WL	$\mathrm{WL}=\overset{N-1}{\underset{n=1}{\sum }}\left|x_{n}-x_{n+1}\right|$
**7**	Envelope	ENV	$\mathrm{ENV}=\sqrt{\frac{1}{N}\overset{N}{\underset{n=1}{\sum }}x_{n}^{2}\;}$
**8**	Total signal energy	Etot	$\mathrm{Etot}=\frac{1}{M-1}\overset{M-1}{\underset{n=1}{\sum }}P_{n}$
**9**	Teager energy in time domain	Ttd	${\mathrm{Ttd}}_{n}=x_{n}^{2}-(x_{n-1}*x_{n+1})$ + window-based moving average
**10**	Teager energy in frequency domain	Tf	$\mathrm{Tf}=\overset{M-1}{\underset{n=1}{\sum }}P_{n}*f_{n}^{2}$
**11**	Modified Teager energy	Tf_mod	$\mathrm{Tf}\_\mathrm{mod}=\overset{M-1}{\underset{n=1}{\sum }}P_{n}*f_{n}$
**12**	Mean of signal frequencies	MNF	$\mathrm{MNF}=\frac{\sum_{n=1}^{M}P_{n}*f_{n}}{\sum_{n=1}^{M}P_{n}}$
**13**	Median of signal frequencies	MDF	$\overset{\mathrm{MDF}}{\underset{n=1}{\sum }}P_{n}=\overset{M}{\underset{n=\mathrm{MDF}}{\sum }}P_{n}$
**14**	Firing rate	FR	$\mathrm{FR}=\overset{M}{\underset{n=1}{\sum }}AP$

**Table 2 sensors-22-05054-t002:** Parameters and the ranges in which they were varied for different algorithms.

	Algorithm	Computational Complexity (O)	Window Size Range	Threshold Range
**1.**	MAV	O(n)	(50–1050) ms	NA
**2.**	Var	O(n)	(50–1050) ms	NA
**3.**	SSC	O(n)	(50–550) ms	0–4 × MAV (rest)
**4.**	ZC	O(n)	(50–550) ms	0–4 × MAV (rest)
**5.**	WA	O(n)	(50–550) ms	0–4 × MAV (rest)
**6.**	WL	O(n)	(50–1050) ms	NA
**7.**	Env	O(n)	(50–1050) ms	NA
**8.**	Etot	O(n × log(n))	(50–1050) ms	NA
**9.**	Ttd	O(n)	(50–1050) ms	NA
**10.**	Tf	O(n × log(n))	(50–1050) ms	NA
**11.**	Tf_mod	O(n × log(n))	(50–1050) ms	NA
**12.**	MNF	O(n × log(n))	(50–1050) ms	NA
**13.**	MDF	O(n × log(n))	(50–1050) ms	NA
**14.**	FR	O(n)	(50–1050) ms	85th–99th quantile

**Table 3 sensors-22-05054-t003:** Computational times for different algorithms across different platforms. Each column except for the last is normalized with respect to the quickest algorithm. The values presented in parenthesis are the result of using CMSIS-DSP functions which were possible only for some platforms. In the case of similar or longer processing times of the CMSIS-DSP compared with native functions, the parenthesized values are omitted. The bottom row contains information about the absolute processing time in the case of a 128-sample-wide window. The last column presents the mean of the individually normalized processing times of each algorithm. Values indicated by bold text are the ones with the best performance.

	Algorithm	PC	Cortex-M7	Cortex-M4	Cortex-M0	ATmega328	Mean
**1.**	MAV	1.4	**1**	**1**	7.8	1.4	2.5
**2.**	Var	1.5	1.5 (1)	1.5 (1)	**1**	**1**	**1.2**
**3.**	SSC	3.3	**1**	1.3	11.3	3.2	4.0
**4.**	ZC	2.9	1.5	1.7	7.9	2	3.2
**5.**	WA	4.7	4 (1.5)	3.3 (1.5)	6.4	1.6	4.2
**6.**	WL	**1**	3 (1.5)	2.7 (1.5)	10.8	2.7	4.8
**7.**	Env	1.1	1.5 (0.5)	**1 (0.8)**	1.2	**1**	**1.1**
**8.**	Etot	5.3	20 (7.5)	170.5 (15.3)	160.5	31	65.6
**9.**	Ttd	1.1	**1**	1.5	23.4	4.3	6.3
**10.**	Tf	5.4	20.5 (9.5)	170.0 (18.2)	162	31.5	66.3
**11.**	Tf_mod	5.4	20 (9.5)	170.0 (18)	161.2	31.6	66.1
**12.**	MNF	5.6	20.5 (9.5)	170.0 (17.7)	167.1	32.1	68.3
**13.**	MDF	16.4	20.5 (9.5)	170.2 (17.2)	182.7	35.1	78.1
**14.**	FR	41.2	3.5	3.0	6.2	1.4	11.1
		1 = 6.9 µs	1 = 2 µs	1 = 6 µs	1 = 0.56 ms	1 = 1.52 ms	

**Table 4 sensors-22-05054-t004:** Statistical significance between algorithms for both RMSE and cross-correlation. Green indicates statistically significant difference (*p* < 0.05; Bonferroni-corrected) for the RMSE metric and orange indicates significant difference (*p* < 0.05; Bonferroni-corrected) for the cross-correlation metric. The cases of significant differences for both metrics are marked with * on the cross-correlation part of the table. Grey indicates diagonal fields.

	MAV	Var	SSC	ZC	WA	WL	Env	Etot	Ttd	Tf	Tf_ mod	MNF	MDF	FR
MAV							*		*			*	*	
Var												*	*	*
SSC												*	*	*
ZC												*	*	*
WA												*	*	*
WL							*		*			*	*	
Env												*	*	*
Etot												*	*	*
Ttd												*	*	*
Tf												*	*	*
Tf_mod												*	*	*
MNF														*
MDF														*
FR														

## Data Availability

The Matlab data container files (*.mat) can be found here: (https://figshare.com/s/06f113bd74ecf6384729 (accessed on 2 June 2022)).

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
