# Peer review of "Evaluation of Simple Algorithms for Proportional Control of Prosthetic Hands Using Intramuscular Electromyography"

_sensors, 2022, doi:10.3390/s22135054_

Round 1

Reviewer 1 Report

The manuscript presents the evaluation of well-known algorithms on iEMG for the control of dexterous myoelectric prostheses. The algorithms, widely applied on sEMG as feature extraction stage, are intended to be used as single commands for a proportional control of a hand prosthesis based on iEMG signals. Intramuscular signals, capable of accurately capturing selective motor units, provide a deeper understanding of muscule contractions in dexterous hand/finger movements against their corresponding muscle force generation. As the authors pointed out, far from computational novelties, the article explores the comparison between a set of methods for implementations of real-time approaches. The manuscript is highly organized, ease to read, includes sufficient details of the methods, and provides a comprehensive result section, with complementary figures in the Appendix section. Also, it includes an extensive discussion about the results and their implications from the point of view of the nature of the signal. For all the above, I think the article could be accept for publication. However, to help to improve the quality of the article, I recommend the following concerns.

Lines 137: Please, indicate to which finger the notations D2 - D5 belong.

Line 151: The Figure reference is wrong.

Line 170: What are the specifications of the band-stop filter used?

The normalization of MAV features presented in Figure 2 was not included in the text of signal pre-processing section. Was this normalization performed for all features as a step of this stage?

Line 376: Was the normal distribution tested for data? Please, include details. 

Line 410: The notation (6 x 16) was not clear .

Figure 7-8: Scales of vertical bars were not clear for both left plots. If they are the same as the right plots, please exclude the bars without the scale (left plot) or include the numerical scale for both plots.   

Author Response

The manuscript presents the evaluation of well-known algorithms on iEMG for the control of dexterous myoelectric prostheses. The algorithms, widely applied on sEMG as feature extraction stage, are intended to be used as single commands for a proportional control of a hand prosthesis based on iEMG signals. Intramuscular signals, capable of accurately capturing selective motor units, provide a deeper understanding of muscule contractions in dexterous hand/finger movements against their corresponding muscle force generation. As the authors pointed out, far from computational novelties, the article explores the comparison between a set of methods for implementations of real-time approaches. The manuscript is highly organized, ease to read, includes sufficient details of the methods, and provides a comprehensive result section, with complementary figures in the Appendix section. Also, it includes an extensive discussion about the results and their implications from the point of view of the nature of the signal. For all the above, I think the article could be accept for publication. However, to help to improve the quality of the article, I recommend the following concerns.

Lines 137: Please, indicate to which finger the notations D2 - D5 belong.

A: We extended the nomenclature to „D2(index finger)-D5(little finger)“ to resolve potential issues.

Line 151: The Figure reference is wrong.

A: We corrected the error.

Line 170: What are the specifications of the band-stop filter used?

A: We added a detailed description of the filter used to remove strong powerline noise.

Line 171-174:

„To remove powerline base frequency (f0) and its harmonics (f0 × i, i=2, 3, 4…) which also exceeded iEMG signal amplitude, a band-stop comb filter com-prising 3rd order Butterworth notch filters (f0=50 Hz and Δf = ±2 Hz) was applied to the recorded signals [27]. “

The normalization of MAV features presented in Figure 2 was not included in the text of signal pre-processing section. Was this normalization performed for all features as a step of this stage?

A: Thank you for the comment. The normalization was part of the „Testing of algorithms stage“ part so we acknowledged this in the figure caption too.

Line 198:

„Figure 2. Schematic representation of the signal pre-processing and testing of algorithms steps“

Line 216-218:

„The performance of algorithms was tested in a similar fashion as shown in Figure 2 by using the optimal pairings between iEMG channels, phases of force channels, and movements found during the signal pre-processing stage.“

Line 376: Was the normal distribution tested for data? Please, include details. 

A: The data was tested for normality using the Lilliefors test which indicated that some of the results were not following normal distribution.

Line 382-385:

“As the data used to compare different algorithms and window sizes did not follow a normal distribution determined by the Lilliefors test, we used the Friedman test with Bonferroni post-hoc correction to assess statistically significant differences of medians.”

Line 410: The notation (6 x 16) was not clear.

A: We included an explanation of this notation to „6 iEMG channels x 16 datasets“

Figure 7-8: Scales of vertical bars were not clear for both left plots. If they are the same as the right plots, please exclude the bars without the scale (left plot) or include the numerical scale for both plots.   

A: We corrected the error with the figure bars.

Reviewer 2 Report

Dear Authors,

I've got only one substantive remark connected with Your work:

in. l. 170-171 there is a statement: "Namely, band-stop filters centered at power line frequency (50 Hz) and its harmonics were applied to the recorded iEMG signals." 

According to my experience with sEMG, You can not use this kind of filtration. This sentence needs a deeper explanation.

My other remarks are:

1. I would recommend correcting the discussion part - to show Your results in the background of other researchers' findings

2. from editorial point of view tables look different (compare tab. 1 and 2), 

3. You are using "*" instead of the multiplication sign

4. l. 36 fingers instead of finger

5. l. 151: (...) that the volunteer produced (see Figure 1Error! Reference source not found.). The idea (...) - see the error problem.

Author Response

Reviewer 2

I've got only one substantive remark connected with Your work:

  1. l. 170-171 there is a statement: "Namely, band-stop filters centered at power line frequency (50 Hz) and its harmonics were applied to the recorded iEMG signals." 

According to my experience with sEMG, You can not use this kind of filtration. This sentence needs a deeper explanation.

A: Thank you for the comment. It is true that for sEMG filtering it is usually not necessary to employ comb filters to remove powerline harmonics. But, in the case of iEMG, the muscle electrical activity is picked by tips of fine-wires that acquire signals from a very small area. Therefore, the amplitude of the recorded iEMG in our database is in the range of 100-200 uV. Furthermore, the fine-wires exit skin and are interfaced with the pre-amplifiers usually distanced >5 cm. These free wires act as antennas, further inducing EM noise in the iEMG signal. As a result, the frequency spectrum of the recorded signals has pronounced 50xN Hz peaks which decrease in size with the higher frequencies. In the case of sEMG, signal strength is usually high enough to prevail over powerline noise harmonics, but, in our case, they were significant even at high frequencies. Therefore, to reduce powerline interference below the spectral components of iEMG, we were forced to construct a comb filter using Butterworth notch filters.

Line 170-174:

“Namely, the powerline noise was superimposed with the iEMG signal which is relatively weak, compared to sEMG. To remove powerline base frequency (f0) and its harmonics (f0 × i, i=2, 3, 4…) which also exceeded iEMG signal amplitude, a band-stop comb filter com-prising 3rd order Butterworth notch filters (f0=50 Hz and Δf = ±2 Hz) was applied to the recorded signals [27].”

My other remarks are:

  1. I would recommend correcting the discussion part - to show Your results in the background of other researchers' findings

A: Thank you for the comment. We included a comparison of our findings with the study done by Kamavuako et al. which is done with a similar protocol as in our paper.

Line 536-563:

“The results of applying simple computational algorithms on the database could be also compared to other studies. In an example, Kamavuako et al. [28] in their study used sine-tracking protocol using isometric forces corresponding to two wrist DoF, which is relatively similar to the protocol used in this paper. The force level was >10% of the MVC level, which is also the case in the study presented here. The results on ten subjects re-ported in Kamavuako et al. show a mean RMSE of ~14% (range between 10% and 19%). These values are in line with the best-performing algorithm (FR) from our study which had a median RMSE of 17.8% (Q1-13%; Q3-26%). For the other algorithms there is partial overlap with the results of Kamavuako et al. study (e.g. for MAV median RMSE is 22.5% with Q1-17%; Q3-29.6%). In terms of cross-correlation metric, Kamavuako et al. achieved a mean Pearson correlation coefficient of ~0.93 (range between 0.88 and 0.96). The results of the presented study show lover cross-correlation between isometric forces and tested sim-ple algorithms. The best-performing algorithm (FR) only partially falls into the range re-ported by Kamavuako et al. with a median cross-correlation of 0.85 (Q1-0.75; Q3-0.92), while for other algorithms, only Q3 values lie above 0.88. We hypothesize that there are three potential causes of somewhat worse results presented in this study compared with the study done by Kamavuako et al. Firstly, the algorithms used in our study are very basic with the normalization of the outputs based on only the first two force sine-tracking repetitions, and no additional optimization was performed. Secondly, the correlations between some of the iEMG channels and the elicited joint forces were labeled as “ques-tionable” during the qualitative evaluation of signal quality done by the neurologist (check [24] for more information). Nevertheless, those pairings were still included in this study as they represent realistic use-cases where fine-wires, or intramuscular electrodes in general, were slightly misaligned with the targeted muscle, or migrated after initial insertion. Thirdly, in the presented study only direct control was considered (one iEMG channel to estimate one joint force) while in the study done by Kamavuako et al., eight iEMG chan-nels were used to estimate two wrist DoF, therefore using more information regarding muscle activity to estimate resulting forces.”

  1. from editorial point of view tables look different (compare tab. 1 and 2), 

A: We changed the outline of Table 1 to match style of the remaining tables.

  1. You are using "*" instead of the multiplication sign

A: Thank you for your comment, we corrected the error related to the multiplication notification.

  1. l. 36 fingers instead of finger

A: We corrected the error.

  1. l. 151: (...) that the volunteer produced (see Figure 1Error! Reference source not found.). The idea (...) - see the error problem.

A: We corrected the error.